# General Strategies for RNA X-ray Crystallography

**DOI:** 10.3390/molecules28052111

**Published:** 2023-02-23

**Authors:** Ryland W. Jackson, Claire M. Smathers, Aaron R. Robart

**Affiliations:** Department of Biochemistry and Molecular Medicine, West Virginia University, Morgantown, WV 20506, USA

**Keywords:** RNA, ribozyme, X-ray crystallography, structure, structural biology

## Abstract

An extremely small proportion of the X-ray crystal structures deposited in the Protein Data Bank are of RNA or RNA–protein complexes. This is due to three main obstacles to the successful determination of RNA structure: (1) low yields of pure, properly folded RNA; (2) difficulty creating crystal contacts due to low sequence diversity; and (3) limited methods for phasing. Various approaches have been developed to address these obstacles, such as native RNA purification, engineered crystallization modules, and incorporation of proteins to assist in phasing. In this review, we will discuss these strategies and provide examples of how they are used in practice.

## 1. Introduction

Of the over 190,000 macromolecular structures present in the Protein Data Bank, as of 2022, only 3678 represent X-ray structures of RNA or RNA–protein complexes. Of these, the majority are of small RNAs (≤50 nucleotides in size), making large RNA crystal structures even more of a minority. RNA crystallography is challenging for several reasons: (1) traditional RNA purification methods resulting in low yields of pure, properly folded RNA; (2) nucleic acids having less sequence diversity than proteins, making crystal contacts harder to come by; and (3) difficulties in solving the phase problem in RNA-only structures.

RNA crystallization is complicated by inherent difficulties, such as vulnerability to degradation by RNases and susceptibility to misfolding. A major bottleneck in RNA crystallography is often the production of sufficient amounts of high-quality, homogeneously folded RNA. This review is meant to be used as a reference for the crystallization of large-structured RNAs such as ribozymes and riboswitches. We will discuss the advantages and disadvantages of both traditional and native RNA purification methods. Furthermore, examples of crystallization strategies using both RNA-dependent and protein-driven modules will be discussed. Finally, strategies for phasing—including molecular replacement and general tools for heavy metal soaking to obtain anomalous diffraction data—will be presented.

## 2. RNA Purification and Folding

T7 RNA polymerase is commonly used to produce large quantities of RNA by in vitro transcription [1]. Despite the versatility of this common method, there are several important caveats to consider. T7 RNA polymerase is prone to non-templated additions of 1–3 nucleotides to the 3′ end of the RNA transcript. These non-templated additions can be circumvented by the incorporation of two sequential 2′-O-methyl substitutions in the last two nucleotides of the 5′ end of the DNA template strand [2]. Template slippage is also known to occur with T7 RNA polymerase when it encounters polyA sequences during transcription [3]. This polymerase also requires the 5′ end of the RNA sequence to contain at least two sequential guanosine residues for efficient promoter firing [1]. Careful sequence design can mitigate these known problems and enhance RNA yield [2]. Heterogeneity in transcript ends can also be overcome using ribozymes.

### 2.1. Producing Homogeneous Transcripts: Hammerhead Ribozyme

In vitro transcription produces a continuum of transcript lengths due to premature polymerase termination. In some cases, this heterogeneity can be problematic for crystallization. To overcome this, self-cleaving ribozymes, such as the hammerhead and *glmS*, can be engineered into the sequence to produce homogenous ends.

The hammerhead ribozyme is a small (~50 nucleotides), self-cleaving ribozyme that is found in plant pathogens [4,5] and in the satellite RNAs of newt mitochondria [6]. The use of the hammerhead ribozyme for directed transcript cleavage proved successful in the crystallization of multiple targets [7,8]. There are two forms of hammerhead ribozyme: minimal and full-length. Both contain the same conserved 13-nucleotide catalytic core, but they differ in catalytic efficiency [9]. Hammerhead ribozymes are optimally active in 10 mM Mg^2+^ at pH 7.5 [10]. Interestingly, the activity of the full-length ribozyme is increased over 100-fold compared to the minimal ribozyme [11]. Implementing hammerhead ribozymes to create homogenous transcript ends requires complementary base pairing with the target RNA (Figure 1A) [12,13]. The hammerhead ribozyme folds into three helices that flank conserved, single-stranded core nucleotides necessary for autocatalytic cleavage (Figure 1A). A conserved group of nucleotides bind Mg^2+^ metal ions required for folding of the ribozyme into its active conformation (Figure 1B) [14,15]. Cleavage occurs when bound Mg^2+^ activates the 2′-hydroxyl of C17 for nucleophilic attack on the adjacent phosphodiester bond between C17 and A1.1, producing a 2′,3′-cyclic phosphate and 5′-hydroxyl termini (Figure 1B) [10]. Use of the hammerhead ribozyme is complicated by sequence requirements immediately upstream of the cleavage site. Mutagenic studies have shown that this sequence usually needs to be an NUH trinucleotide (N = any nucleotide, H = not G), but that CAC, CGC, and AAC trinucleotide sequences also lead to efficient cleavage [16,17,18,19].

### 2.2. Producing Homogeneous Transcripts: glmS Ribozyme

The *glmS* ribozyme is a self-cleaving sequence found in the 5′ UTR of the *glmS* gene, encoding the protein glucosamine-6-phosphate aminotransferase. In vivo, this ribozyme serves as a riboswitch, regulating glucosamine-6-phosphate (GlcN6P) production by inactivating *glmS* mRNA through self-cleavage only when GlcN6P is bound in the active site [20,21]. The *glmS* ribozyme consists of a nested double-pseudoknot fold at its core, and another pseudoknot-containing domain peripheral to the core (Figure 1C) [20,21,22]. Pre-catalytic *glmS* ribozyme structures from *Bacillus anthracis* confirmed that effector binding immediately initiates the cleavage reaction through acid–base catalysis (Figure 1D) [20,22]. The *glmS* ribozyme requires both the GlcN6P effector and the G33 base for activity, and mutation of G33 disrupts catalytic activity even though it does not affect ribozyme folding [22].The self-cleavage activity of the *glmS* ribozyme can be utilized to produce homogeneous transcript ends through incorporation of the ribozyme at the 3′ end of the target RNA sequence. Addition of GlcN6P induces site-specific self-cleavage of the *glmS* ribozyme, producing the co-transcriptionally folded target RNA with homogenous 3′ ends [23].

### 2.3. Purification of Transcribed RNA

Properly folded RNA is critical, as conformational heterogeneity leads to poorly ordered crystals and an inability to diffract at a sufficient resolution. Purification of transcribed RNA has been accomplished by excising bands from denaturing PAGE gels, followed by a refolding protocol where the RNA is heat-denatured at 90 °C and allowed to refold by gentle cooling [23]. Unfortunately, temperature-based folding methods have proven unsuccessful for many RNAs due to misfolding. For example, Pereira et al. showed that heat refolding of the VS ribozyme after T7 transcription resulted in conformational heterogeneity and an inactive ribozyme. Urea titrations in conjunction with the same denaturing protocol can be used to mitigate this [24,25,26,27]. Moreover, the small amount of UV used to visualize RNA on a polyacrylamide gel can permanently damage the sample [28]. RNA itself has a free energy of folding of over 100 kcal/mol, so complete denaturation may never fully occur before refolding occurs [29]. Stacking interactions and hydrogen bonding between the small number of bases that comprise RNA are far more prevalent than niche long-range tertiary interactions, which create many different conformational folding possibilities rather than a select few unique folds. In addition, PAGE purification leads to residual acrylamide oligomer contamination that binds to the RNA and is impossible to eliminate, leading to deceptive increases in the molecular weight of the RNA and loss of significant amounts of transcriptional yield from the formation of irreversible aggregates caused by partial denaturation [30]. These issues can make obtaining the high yields needed for crystallography difficult.

Native folding of RNA during transcription can eliminate almost all of the aforementioned issues. Larger RNAs, such as group II self-splicing introns in both pre- and post-catalytic states [31,32], demonstrate a difficulty in maintaining an active fold. Using native gel analysis, Toor et al. showed that denaturing methods prevent refolding from occurring. To circumvent this, a large group of intron-variant sequences were screened to try to find one that had high splicing activity in low-magnesium and higher-temperature conditions [31]. Once the effective intron sequence was found, it was allowed to fold co-transcriptionally using T7 RNA polymerase. Next, the mixture was treated with DNase I to remove the residual DNA template, and then with Proteinase K to remove the DNase and T7 RNA polymerase. Finally, the Proteinase K (29 kDa) was removed by ultrafiltration using an Amicon 100 kDa cutoff filter. During this process, many washes with a simple buffer containing MgCl_2_ and sodium cacodylate (pH 6.5) were added, allowing buffer exchange and concentration of the RNA [31]. This process leaves fully folded and active catalytic RNA without contaminants.

RNAs natively fold when transcribed in cells and can possess unique post-transcriptional modifications that are important for understanding their structure and function. Modifications can include methylation [33], acetylation [34], glycosylation [35], and many others. In these cases, in vitro transcription is not feasible and other methods must be implemented. Overexpression of a vector containing the target RNA, an inducible promotor, and an affinity purification tag is one way to create modified RNA targets. The viral coat protein MS2 is a small RNA-binding protein [36] that has been widely used as a tool for purifying RNA. This protein binds specifically to a hairpin RNA motif called the MS2 aptamer. The aptamer can be engineered peripherally to the RNA of interest or within non-essential solvent-exposed structured RNA regions. By fusing the MS2 protein to a purification tag such as a FLAG-, HA-, or GST-tag, it is possible to selectively purify the RNA target using affinity chromatography [37]. Other RNA-binding proteins, such as *Pseudomonas* phage 7 (PP7) [38], can also be used and can function to purify RNA as well as RNA–protein complexes [39,40].

## 3. RNA-Driven Crystallization Modules

The crystallization of large RNAs can be a difficult task, as nucleic acids do not tend to crystallize as readily as proteins. The diversity of protein side chains offers more opportunities for crystal contacts due to properties such as differential charge, polarity, and hydrophobicity [25,41]. In contrast, nucleic acids present a continuous surface of negative charge that may serve to repel other molecules and inhibit crystal contact formation [25]. Luckily, strategies exist to engineer RNAs containing secondary structure domain modules that can promote crystallization.

Placement of engineered crystallization modules requires careful consideration to ensure solvent accessibility and to prevent the disruption of the native core RNA structure. A useful tool to help guide the modification of the RNA crystallization target is the use of phylogenetic sequence analysis to identify non-conserved variable regions within the RNA periphery. These variable regions are most likely non-essential to the function of the RNA and, thus, can be altered in order to promote crystallization. It is important to have an experimental method to assess the activity and proper folding of the modified RNA sequence following the addition of such modules. Information on phylogenetic variants can be obtained from RNA family databases such as Rfam [42]. Additionally, small changes in helical lengths and terminal loops can have a significant effect on the formation of lattice contacts [43]. Thus, several variations of the engineered molecule, such as the addition or subtraction of base pairs, should be screened for crystallization and diffraction. The formation of RNA’s tertiary structure relies primarily on interactions between secondary structure elements. Figure 2A shows variable regions identified through phylogenic analysis of group IIC introns that could be altered without disrupting folding or activity. Altering of the length of peripheral helices can be particularly helpful in changing how crystal contacts form between molecules and, thus, may aid in the formation of well-ordered crystals.

### 3.1. Tetraloop Interactions as RNA Crystallization Modules

Crystallization of large RNAs often requires the insertion of a “crystallization module”, which promotes nucleation and enhances crystal growth. This method takes advantage of common RNA tertiary structural elements such as tetraloop (four-nucleotide loop sequence) [44] interactions and kissing loop interactions [45]. Tetraloops are common hairpin loop motifs found in RNAs [46], with GNRA (N = any nucleotide, R = A or G) tetraloops being the most prominent in naturally occurring folded RNAs [44]. The GAAA tetraloop is frequently used to enhance crystal contacts and has been used in the contexts of both random and specific tetraloop receptor binding [31,32,47,48,49,50,51]. Crystallization of the SAM-I riboswitch [52]—an mRNA element that binds S-adenosyl methionine to regulate gene expression in bacteria [53,54]—is one example of the GAAA tetraloop used for random binding in conjunction with peripheral helical length variation. GAAA tetraloops tend to bind tandem GC pairs in minor grooves of RNA helices [55,56] or sequences consisting of two Watson–Crick GC pairs, a reverse Hoogsteen AU pair, an adenosine platform, and a wobble GU pair [57]. However, biochemical studies have shown that both the GNRA tetraloops and the receptor sites can tolerate a high degree of variability without losing their binding affinity or specificity [58,59].

The interaction between a GAAA tetraloop and a specific engineered tetraloop receptor can also be used as a crystallization module. The GAAA tetraloop, in conjunction with the 11-nucleotide tetraloop receptor motif (Figure 2B), has been extensively studied and utilized for this purpose. Without affecting the structure biochemically, this module has been used to crystallize multiple large RNAs, including domains 5 and 6 of the group IIB intron ai5γ [47], the human hepatitis delta virus (HDV) ribozyme [47], a CUG RNA helix implicated in myotonic dystrophy type 1 [48], a bacterial ribonuclease P holoenzyme in complex with tRNA [49], and four separate group II intron structures (Figure 2C) [31,32,50,51]. This interaction was first identified by Murphy and Cech, who observed a tertiary contact between a GAAA tetraloop and a conserved bulge on a distal helix stabilizing the fold of the *Tetrahymena thermophila* ribozyme’s P4-P6 domain [60]. Costa and Michel later characterized the GAAA-tetraloop-specific 11-nucleotide receptor as a highly conserved asymmetric internal loop with the sequence 5′-UAUGG-3′:5′-CCUAAG-3′ [57]. Finally, the structure was obtained by Cate et al. [61]. This interaction is a good choice for enhancing crystallization because of its strength and specificity, acting as a thermodynamic clamp [60,62]. GAAA tetraloop–tetraloop receptor interactions have also been shown to positively affect the accuracy of ribozyme folding pathways [63,64,65,66,67] and, when disrupted by mutation, can cause destabilization of other tertiary interactions within the folded RNA structure [64].

### 3.2. Loop–Loop Interactions as RNA Crystallization Modules

As mentioned above, loop–loop or “kissing” loop interactions are another RNA tertiary motif that can be utilized as a module for crystallizing large RNAs [50,68]. Kissing loop complexes are formed by base pairing between the single-stranded residues of sequence-complementary loops [45]. Restricted forms of intramolecular kissing loop interactions were first identified between the D and T loops of tRNAs [69], and more extensive interactions were later found in the peripheral components of subgroup IC1 and ID introns [70]. Here, loop residues participate in intramolecular base pairing, creating a single composite, coaxially stacked helix composed of the two original hairpin loops and a new helix created by Watson–Crick base pairing of the nucleotides in the complimentary loops between the two original stems [71]. All nucleotides in each loop are stacked on the 3′ side of the main helix and are involved in pairing interactions [71]. Overall, the structure of the interaction resembles a bent RNA helix and requires magnesium ions to form [72]. Kissing loop interactions can also be intermolecular, which has been observed in the recognition of complementary anticodons between different tRNA pairs [73,74,75,76,77,78], the dimerization of genomic RNA of retroviruses [79,80], and in reverse transcription during HIV-1 replication [81].

The most notable example of a kissing loop interaction being used as an RNA crystallization module is in the human spliceosomal U1 snRNP structure [68]. The Nagai lab set out to crystallize the full U1 snRNP complex in 2009 and made many changes to the sequences of both the RNA and proteins to obtain constructs that would support crystallization. Previous studies had shown that the U1A binding site of the U1 snRNA is not crucial to U1 snRNP activity, making the region containing this sequence a reasonable place for the insertion of a kissing loop interaction in the crystallization construct [68]. After soaking with heavy metals (see below), the team obtained crystals that diffracted to 5.5 angstrom [68].

## 4. Protein-Assisted RNA Crystallography

Crystallization modules can also be composed of RNA-binding proteins or individual protein domains. Protein crystallization modules introduce surfaces that are chemically different from the negatively charged RNA surface, which help to position molecules in a repeating pattern and facilitate the growth of well-ordered crystals [41].

### 4.1. U1A Protein Module

The most widely used protein crystallization module has been the U1A protein—one of the components of the spliceosomal U1 small nuclear RNP (snRNP), which specifically recognizes a 10-nucleotide sequence in stem-loop II of the U1 snRNA [41]. This sequence can be engineered into a functionally unimportant stem-loop of the RNA target to facilitate binding to the RNA recognition motif (RRM) of the U1A protein and co-crystallization of the resulting RNP [41]. The crystal structure of the U1A protein bound to its cognate RNA shows that the RNA–protein interactions are confined to only seven nucleotides at the 5’ end of the 10-nucleotide loop and the closing base pair of the stem (Figure 3A) [82]. This makes insertion of a U1A binding site into an RNA target quite simple, requiring only a 12-nucleotide insertion to recapitulate the site [82]. Several RNA structures have been solved using the U1A crystallization module, including the hairpin ribozyme [83], the *glmS* ribozyme–riboswitch [84], the *Azoarcus* group I intron [85], and the HDV ribozyme [86]. This module has also been used for the crystallization of in vitro evolved ribozymes, aptamer domains, and artificial riboswitches [87,88,89,90].

### 4.2. Kink-Turn Module

Another protein crystallization module used to create crystal contacts is the kink-turn (k-turn) motif. This RNA motif, originally discovered in 2001 by Klein et al. in the 50S ribosomal subunit of *Haloarcula marismortui*, is ~15 nucleotides, containing two helices interrupted by a 3-nucleotide bulge [91]. The asymmetric bulge bends the helical axes 120°, leaving the three unpaired nucleotides free to interact with a k-turn binding protein [91,92]. Flanking the 5′ side of the bulge is a canonical (C) helix, containing Watson–Crick base pairs, and a 3′ non-canonical (NC) helix that begins with 2-3 G-A base pairs (Figure 3B) [91,92].

The interactions between k-turns and the L7Ae family of proteins are similar to those of DNA-binding proteins [92,93]. The alpha helix enters the major groove, made possible due to the kinked shape [92,93]. There, it interacts both nonspecifically with the RNA backbone and specifically via hydrogen bonding with the guanine bases in the NC helix (Figure 3B) [91,92,93]. A hydrophobic loop in the L7Ae protein also interacts with the unpaired bases in the kink itself [91,92,93]. Together, these interactions have a binding affinity of around 10 picomolar [93]. The k-turn motif, in conjunction with the bacterial L7Ae family protein YbxF, has been used to facilitate the crystallization of a T-box riboswitch stem I domain in complex with its cognate tRNA [94]. The k-turn RNA–protein complex facilitated crystal contacts and provided phase information (see below) [94]. The k-turn motif is a useful tool to co-crystallize RNA because it can easily be added to peripheral areas of RNA (Figure 3B), and the abundance of proteins that bind the motif provides many options for co-crystallization and is known to affect the packing of the crystals [95].

### 4.3. Antibody Fragment Module

Antibody fragments have been used as crystallization modules or chaperones for many proteins that have proven difficult to crystallize under traditional methods [96,97,98,99,100]. In the last 10 years, antibody fragments (Fabs) have also been developed as a crystallization module for RNA. Fabs provide a large surface area for promoting crystal contacts, primarily through their beta-rich secondary structures, which can also serve as molecular replacement search models [96]. An advantage of using this module for crystallization is that the RNA does not need to be engineered, and structures of the RNA–protein complexes can be determined from the natively folded RNA. One hurdle in developing Fab crystallization modules is that RNA does not trigger the production of antibodies when introduced into an animal system [101]. Thus, a synthetic method must be applied. To produce Fabs that bind to RNA with high affinity, Piccirilli and Koldobskaya developed an M13 phage display platform to present Fab fragment libraries fused to coat proteins [102,103]. Through multiple rounds of selection, Fabs that bind the target RNA can be selected and enriched [101,104]. Hydroxyl radical protection assays performed in the presence and absence of the Fab can identify the epitope recognized on the RNA and determine whether binding disturbs the global fold [101]. This approach has been used in the crystallization of the group I intron P4-P6 domain [104] and an in vitro evolved class I RNA ligase ribozyme (Figure 3C) [90,101].

## 5. Solving the “Phase Problem” for RNA Crystals

Formation of electron density maps of crystallized macromolecules requires the amplitude and phase of each diffracted wave [105]. X-ray diffraction datasets collected from a crystal use predetermined X-ray energies, and the intensities of diffracted waves—or “diffraction spots”—are used to determine amplitude [105]. However, this information is essentially useless without a means to determine the phase of each wave. Phase information is necessary to offset the scattered waves when they are added together during reconstruction of the electron density map; consequently, they are critical for building structures from diffraction data [106]. Unfortunately, unlike amplitudes that can be directly measured as intensities on an X-ray detector, information regarding the phase is lost and cannot be directly observed without specific additional experimental considerations [105,106]. This inherent block between crystal diffraction data and a solved structure is referred to as the “phase problem” in crystallography. Several strategies for solving the phase problem have been developed, such as molecular replacement, various methods of isomorphous replacement, and anomalous diffraction [105,106,107]. In this section, we will discuss molecular replacement models for RNA and RNA–protein complexes, as well as isomorphous replacement with multiple different heavy metals used to support anomalous diffraction methods.

### 5.1. Molecular Replacement Methods

Molecular replacement (MR) is one commonly used method for solving the phase problem, especially for protein crystallography [105,106,107]. This method applies the phases of a structurally similar model to the experimental diffraction data of the target crystal in order to obtain preliminary electron density maps [105,106,107]. Although MR is applicable for estimating phases for any type of macromolecule, it is often better suited for proteins or nucleic acid–protein complexes, as nucleic acid structures make up only 1.8% of the total number of structures in the PDB [108]. This dearth of solved nucleic acid structures can make finding a suitable model for molecular replacement of an all-nucleic acid target quite difficult. MR search models should have high structural similarity to the target molecule [107]. If possible, the input of weak experimental phases determined by anomalous scattering into the search for a model will enhance the chances of success [109]. For a comprehensive review on finding MR models for RNA, see [108].

If no structures are available, a search model can also be designed by homology modeling of the target molecule, or by de novo structure predictions [110,111]. Computational structural biology powered by artificial intelligence (AI) has been revolutionary, providing powerful tools to model macromolecular structures and predict their functions. Alphafold2 is an AI system that uses deep learning algorithms to predict protein structures with astonishing accuracy and is a promising prospect for MR phasing [112]. RNA-based prediction algorithms are also being developed [113,114]: the Rosetta framework FARFAR2 (fragment assembly of RNA with full-atom refinement) uses small RNA fragments that are mended together to create predictions, and it generally performs well in recovering known native-like structures of RNA [114]. Though these advanced computational methods can be limited in the size of the macromolecules that they can predict, their potential applications in phasing will enable a robust and potentially automated pipeline to solve the phase problem in crystallography.

Molecular replacement can be a particularly convenient method for phasing RNA–protein complex crystals, especially when the protein in the complex has already been solved. For example, protein crystallization modules such as U1A [41], L7Ae family proteins [92], and Fabs [90,101] can serve as MR search models. Additionally, known RNA structures have been used, such as with a T-box leader RNA in complex with tRNA, where an existing tRNA structure was used as the search model to solve the phase of the RNA complex [115,116]. Individual homologous domains or subdomains consisting of short helical fragments can also be used [108]. The solved structures of proteins can also be used in conjunction with small RNA fragments designed to find partial MR solutions, which have been used to solve the structure of the flexizyme [105,117] and the c-di-GMP riboswitch [88].

In addition to serving as a search module in molecular replacement methods, methionine residues in protein crystallization modules can also be replaced with selenomethionine derivatives. Selenomethionine substitution of Met sites has shown to make the U1A module suitable for phase determination by multiwavelength anomalous dispersion (MAD; see below) [47,118]. This module was used in the determination of the hairpin ribozyme structure [119], where co-crystals containing selenomethionyl U1A grew readily under the same crystallization conditions as methionine-containing U1A co-crystals. This strategy has also been implemented in the co-crystallization of stem I of the T-box riboswitch that bound a selenomethionyl YbxF [94].

### 5.2. Isomorphous Replacement and Anomalous Scattering

Most large RNAs bind metal ions such as Mg^2+^ or Mn^2+^ that support both their structural integrity and catalytic activities [120,121]. In isomorphous replacement (IR) phasing, the native metal ions in the crystal are replaced with heavy metal ions [105,106]. This substitution results in a heavy-atom derivative crystal that shows measurable scattering intensity differences compared to the native crystal. The scattering intensity difference can then be used to determine the heavy atom positions and phases, allowing the phase of the native RNA structure to be calculated [105,106]. The IR method hinges entirely on the ability to create heavy-atom derivative crystals that are isomorphous with the natural crystal, meaning that they have the same unit cell and orientation of the molecule within the cell [105,106].

Isomorphous replacement can be performed as a single method but has often been combined with anomalous scattering [105,106]. Here, the X-ray energy is tuned to the absorption edge of the IR heavy metal, promoting excitation of inner-shell electrons [105,106]. There are two types of anomalous scattering experiments: multiwavelength anomalous diffraction (MAD), and single-wavelength anomalous diffraction (SAD). With MAD, data are collected from a single crystal at several wavelengths (usually three) to maximize absorption and anomalous diffraction. Wavelengths are chosen at the IR metal’s absorption peak, point of inflection, and at a remote point on the absorption curve of the heavy metal used for phasing [106]. SAD is measured only at the absorption edge peak and is still subject to phase ambiguity [106].

#### 5.2.1. Isomorphous Replacement with Mg^2+^ Mimics

Heavy-atom derivatives are typically produced using metals that mimic Mg^2+^’s binding to RNA [105]. Mg^2+^ is crucial to the structure and folding of RNA and is frequently found coordinated to the negatively charged phosphate backbone or in the major groove at the base edge of tandem guanines [121]. The Mg^2+^ ion prefers octahedral geometry of coordination and can adopt a fully hydrated coordination sphere, Mg(H_2_O)_6_^2+^, or a partially hydrated shell in which inner-sphere contacts are provided by the RNA [121]. Heavy metals that mimic the RNA binding of either the fully hydrated sphere (outer shell) or the partially hydrated shell (inner shell) Mg^2+^ have been used extensively for phasing RNA crystals, as they tend not to disrupt the structure [50,122,123,124]. Hexamine salts have been the predominant ions used as Mg(H_2_O)_6_^2+^ mimics, including those of Co(III), Os(III), and Ir(III) [124]. Each of these ions adopts strict octahedral coordination geometry and exhibits nearly the same coordination distance between the ion and the amine as between Mg^2+^ and water [122]. Hexamine complexes tend to bind RNA almost exclusively through outer-shell contacts. This is because the NH_3_ group is unable to accept a hydrogen bond, unlike H_2_O, which means that the amine coordination shell will gravitate to negatively charged environments [122]. Amine groups within the coordination sphere of Co(NH_3_)_6_(III) also resist exchange relative to the rapid exchange observed for water in the coordination sphere of Mg(H_2_O)_6_^2+^ [125]. Iridium and cobalt(III) hexamine salts are relatively easy to produce in the lab and have been used for phasing of RNA structures such as the 70S ribosome functional complex [126], the P4-P6 group I ribozyme domain [61] and, more recently, a group IIB intron lariat [50].

Inner-shell Mg^2+^ mimics typically include heavy metals in the lanthanide series, such as Yb^3+^, Sm^3+^, Ln^3+^, and Eu^3+^. Diffraction experiments on the P4-P6 group I ribozyme domain led to the observation that the unit-cell dimensions changed as a function of increasing ionic radius for lanthanides in the series from Lu^3+^ to Sm^3+^, and that the mosaic spread of the diffraction pattern increased as a function of increasing ionic radius for all lanthanides except Sm^3+^ [127]. Inner-sphere contacts tend to be catalytically important molecules and are more rarely found in RNA structures. Lanthanide metals have been used to successfully phase crystals of multiple RNAs such as tRNA [128,129], a hammerhead ribozyme [56], group II introns [31,50], and the *Azoarcus* group I ribozyme [85].

#### 5.2.2. Engineering Heavy Metal Binding Sites

Heavy metal derivatives have historically been produced by a method affectionately referred to as “soak and pray”, where the crystal is soaked in a heavy metal atom solution with the hope that the heavy metal atoms will bind to one or more specific locations within the RNA [124]. Although this method typically results in derivatized crystals, RNA containing suitable specific sites for heavy metal binding is not predictable; thus, the process becomes highly time-consuming through rounds of trial and error [130,131,132]. To address this issue, a general “directed soaking” method has been devised by Batey and Kieft that involves engineering one or more reliable, non-structure-perturbing cation-binding sites into the RNA structure and then soaking hexamine cations into resulting RNA crystals [123,124]. Their method is based on the observation that G-U wobble pairs in A-form helices create a binding site for many cations, including hexamine complexes [127,133,134,135,136]. The identity and orientation of the base pairs that flank the wobble pairs should be taken into consideration, as this can affect cation binding [124]. This method has successfully been used with both cobalt(III) and iridium(III) hexamine, but even cesium has been shown to be effective for phasing when bound to the motif [137]. Since engineering of this site only changes a few nucleotides, it can typically be performed without perturbing the fold or function of the RNA.

#### 5.2.3. Incorporation of Selenium into Nucleic Acids

X-ray crystallography of proteins has been greatly impacted by the utilization of selenium derivatization as a phasing module [138]. Selenium is a popular choice for phasing by anomalous diffraction and is in the same periodic family as oxygen and sulfur; thus, selenium substitution often does not cause structural perturbations [138]. Although this method typically involves substituting selenium for sulfur atoms in methionine residues, selenium has been successfully incorporated into large nucleic acids via multiple enzymatic approaches [138,139]. Small selenium-derivatized nucleic acids (60 nt. or less) can be produced easily during oligonucleotide synthesis, while larger ones can be prepared using DNA or RNA nucleotide triphosphates ((d)NTPs) incorporated by polymerase activity [138]. These Se-modified dNTP/NTPs are commercially available and can include selenium substitutions in the base, sugar, or phosphate portions. For the preparation of large selenium-derivatized RNAs, in vitro transcription using NTPαSe analogs has been used, where one of the oxygen atoms on the alpha phosphate of the NTP is replaced with selenium [138]. These analogs perform as substrates for T7 RNA polymerase just as well as natural NTPs, but certain bases may affect the activity of the resulting Se-modified RNAs. When in vitro transcriptional incorporation of NTPαSe analogs into the hammerhead ribozyme was tested, it was observed that ribozymes produced with UTPαSe and CTPαSe analogs had the same catalytic activity as the wild type [138]. However, ribozymes produced with GTPαSe had only 30% wild-type activity, and ribozymes produced with ATPαSe had very low activity [138]. This suggests that the incorporation of selenium into an RNA crystallization target may require some trial-and-error optimization. Aside from direct polymerization incorporation, Se-modified RNAs may also be produced by enzymatic ligation of two or more selenium-containing fragments. This method was used in the crystallization of a rat spliceosomal U6 snRNA stem-loop motif using T4 RNA ligase [140,141].

#### 5.2.4. Soaking with Halogens

MAD and SAD methods combined with halogen soaking have also been used as techniques for phasing, with limited success [142]. Both bromine and iodine have been utilized, where the halides incorporate into the ordered solvent shell as anomalous scatterers. The absorption edge of bromine is achievable at all synchrotron beam lines and, although the absorption edges of iodine are not easily accessible, it does have a significant anomalous effect [142]. Soaking with this method can require high halide concentrations (0.2–1 M), and soaks should last only a few seconds because of their fast diffusion into the crystals [142].

## 6. Conclusions

X-ray crystallographic methods have proven to be an invaluable tool in the study of large RNAs. The purification, crystallization, and phasing strategies presented here have helped counter the inherent challenges of RNA crystallography, enabling the determination of structures of many difficult-to-crystallize RNAs. The development of new tools and techniques is continuing to improve the convenience and resolution of these methods, allowing us to understand the functions and potential applications of structured RNA in research and therapeutics. In addition, cryo-electron microscopy (cryo-EM) is gaining traction as a method of RNA structural biology. It is likely that combined approaches using X-ray crystallography, cryo-EM, and even small-angle X-ray scattering (SAXS) will become increasingly popular to fully appreciate the dynamics of structured RNA molecules and to validate structural observations.

## Figures and Tables

**Figure 1 molecules-28-02111-f001:**
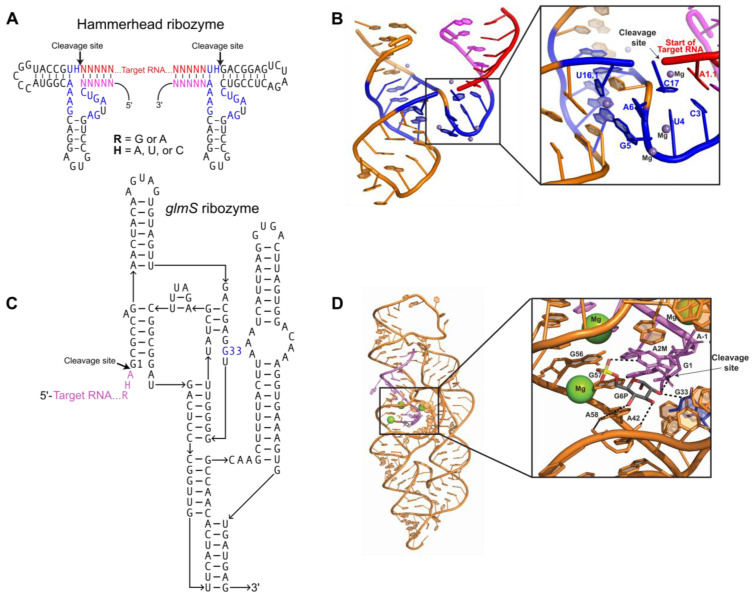
Ribozymes for homogenous RNA end production: (**A**) Secondary structure of two hammerhead ribozymes connected by an RNA crystallization target. Conserved nucleotides necessary for catalysis (blue); short complements (magenta) to the 5′ and 3′ ends of the target RNA (red) must be engineered into the ribozyme sequence. Canonical base pairs in helical stems are denoted by dashes. R = G or A; H = A, U, C; N = A, U, C, G. (**B**) Tertiary structure of the minimal hammerhead ribozyme (PDB:300D). Color scheme matches Figure 1A. Inset shows the ribozyme active site turned upward, with catalytic residues, participating substrate residues, and metal ions labeled. (**C**) Secondary structure for a *glmS* ribozyme consisting of the target RNA (purple) and ribozyme strand (black). The G33 (blue) residue interacts with Glc6P and is required for catalysis. (**D**) Tertiary structure of the *Bacillus anthracis glmS* ribozyme (PDB:3L3C) bound to Glc6P (shown as sticks) before cleavage. Color scheme matches Figure 1C. Inset shows the ribozyme active site, with catalytic residues and participating substrate residues (shown as sticks) labeled. Hydrogen bonds between catalytic residues, substrate, and Glc6P indicated by black dashes; cleavage site indicated by arrow. Conserved G33 residue colored blue; bound Mg^2+^ shown as green spheres.

**Figure 2 molecules-28-02111-f002:**
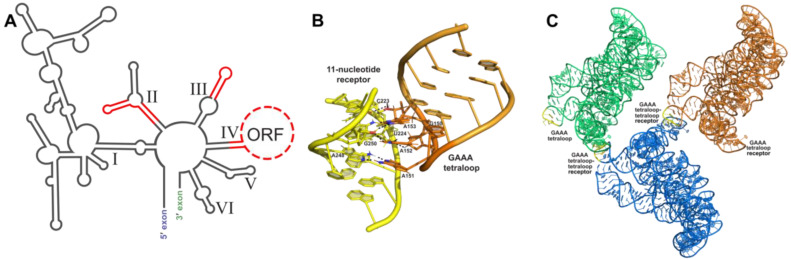
Tetraloop–tetraloop receptor crystallization module: (**A**) Phylogenetic covariation in group IIC introns. Stems and loops in red are variable regions and are amenable areas to engineer crystallization modules. (**B**) Interaction between the GAAA tetraloop (orange) and the 11 nt tetraloop receptor (yellow) in the *O.i.* group II intron (PDB:3IGI); hydrogen bonding shown by black dashes. (**C**) Crystal symmetry present in one unit cell from the *O.i.* group II intron crystal structure (PDB:3IGI). Three RNA molecules pack into one unit cell, each one colored a different color. GAAA tetraloops are colored yellow to show where they form crystal contacts between RNA molecules.

**Figure 3 molecules-28-02111-f003:**
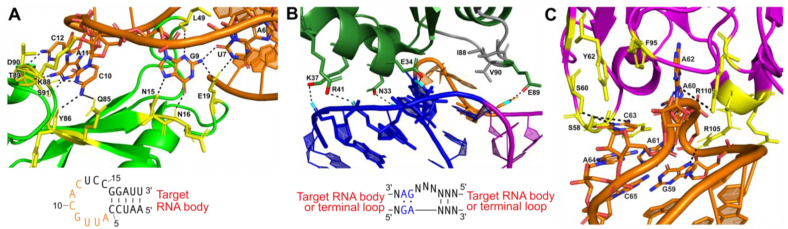
Protein-assisted crystallization modules: (**A**) Interaction between U1A (green) and a 21-nucleotide (nt) hairpin (orange) (PDB:1URN). Several residues form hydrogen bonds (black dashes) with 7 nucleotides of the 12 nt stem-loop U1A binding site in the RNA hairpin. The interaction is further stabilized by base stacking between adjacent nucleotides in the stem-loop. Participating amino acid residues are colored yellow. The hairpin sequence for engineering is also shown with interacting nucleotides (orange). (**B**) Interaction between ribosomal protein L7Ae and a kink-turn (PDB:4BW0). The NC helix (blue) and C helix (magenta) are shown. The bulge (orange) forms hydrophobic interactions with several residues in L7Ae (green; hydrophobic interacting gray). The consensus sequence for engineering is also shown. (**C**) Interaction between the Fab BL3 antibody (purple) and the GAAACAC stem-loop binding site (orange) in an in vitro evolved RNA ligase (PDB:3IVK). Several base stacking interactions and hydrogen bonds (black dashes) form between nucleotides of the stem-loop and amino acids from the Fab. Base stacking between nucleotides of the stem-loop also occurs. Participating amino acid residues are colored yellow. No engineering sequence is shown, as antibody–RNA interactions are unique for each target obtained from the phage display pool.

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
