# Peer review of "General Strategies for RNA X-ray Crystallography"

_molecules, 2023, doi:10.3390/molecules28052111_

Round 1

Author Response

We thank the reviewer for their thorough reading of the manuscript and for their insightful comments and suggestions.  We have addressed these points as described below, and feel that the manuscript has been improved by their incorporation.

Responses to reviewer 1 comments:

  1. The manuscript mainly focused on the crystallization scenarios of RBP with unmodified  RNA and totally ignored modified RNA molecules, which does not reflect the real-life  world. All types of RNA are heavily modified, and these modifications are important to  their foldings, stabilities and functions, etc. Although people rely on in-vitro transcription  to obtain RNA of large quantities, they also obtain modified RNA through various means  like overexpression. As a comprehensive review, this part should not be overlooked and  helps put it into perspectives.  

We thank reviewer 1 for this perspective.  We have added a short section regarding the purification of RNAs transcribed via over expression and purification through popular techniques such as MS2 and PP7 pull-down (lines 147-160).

  1. Figure 3A and 3C: the residues and nucleotides in both figures are not or only partially  labeled, and too many details are shown. The authors are suggested to emphasize the key  contacts and information they want to present. 

We have reworked Figures 3A and C to be less busy and easier to understand and included secondary structure information to help guide interpretation where possible.

  1. Section 5.1-Molecular Replacement Methods: the MR method can also be used for RNA  with known general shapes like tRNA molecules crystallized either alone or in a protein RNA complex; Section 5.2-Isomorphous Replacement and Anomalous Scattering: here the  authors failed to mention the iodine phasing through quick soaking, which would give  excellent anomalous diffraction signals by a home X-ray source. 

We have included more information and examples about use of RNA structures in molecular replacement (line 396-400) and have added a short section about quick halide soaks for phasing (line 515-523).

  1. The manuscript described many well-studied/established examples like the glmS ribozyme-riboswitch, the group I intron and the HDV ribozyme, etc. These examples are probably already familiar to the readers. The authors are encouraged to describe some more protein-RNA interaction cases reported in recent years like the Crispr-cas systems, RNA-sensing modules during viral infections, which are hot topics and elucidated by crystallography. Additionally, computational structural biology also attracts a lot of attention recently, and algorithms like alphfold2 or ResettaFold helps tremendously in crystallography phasing. Many programs predicting protein-RNA complex structures based on ab-initio calculation or homology modeling can be found in the literature as well. The readers would greatly benefit from the discussion of these new things.

We thank the reviewer for their suggestion to include extra RNA-protein interactions, but as the review is meant to be used as a general guide for crystallizing RNA, these interactions are limited in their ability to be used as general modules for various RNA targets. However, we agree that computational structural biology is an important aspect and hot current topic to add when talking about finding and creating MR search models and have added a short section about this technology and its prospects (line 382-392).

Reviewer 2 Report

Determinations of the crystal structures of large RNA and its protein complex are difficult due to difficulties in preparing pure and properly folded sample, the crystallization due to low chemical diversity, and the phase determination. The review well summarizes the recent progress in overcoming these difficulties. Enzymatically produced RNA have heterogeneity due to non-templated addition and template slippage. Introduction of hammerhead ribozyme or glmS ribozyme overcomes this problem. Common purification methods using denatured PAGE introduce conformational heterogeneity. UV irradiation to detect RNA also introduces conformational heterogeneity. Native folding of RNA during transcription can overcome these problems. Enhancement of crystallization possibility can be accomplished by the introductions of artificial RNA-RNA interaction or protein-assisted crystallization. For solving phasing problem, molecular replacement method is applicable for RNA-protein complex. Because RNA can bind Mg2+ or Mn2+ ions, isomorphous replacement or phasing techniques using anomalous diffraction are applicable if these ions can be replaced by other heavy metals such as Co3+, Os3+, and Ir3+. Selenium incorporation into RNA is also a promising method to solve phasing problem. The review is suitable for publication in Molecules if some minor comments shown below are incorporated.

Minor comments

1.    The figure presented in the manuscript is incorrectly numbered. Section 4.2 describes kink-turn module, but the figure number shown at lines 280, 285 and 292 is Figure 3C which represents the complex with Fab antibody. These should be Figure 3B. Figure 3C seems to be for section 4.3. The crystal structure represented in Figure 3C seems to be cited in Reference 81 according to the PDB, but Reference 81 does not appear in Section 4.3.

2.    References 53 through 115 seem to be off by one because the reference following “Cate et al.” in line 206 should be the reference 114. Please revise correctly.

3.    In line 401, it is better to insert “anomalous” before diffraction.

Author Response

We thank the reviewer for their thorough reading of the manuscript and for their insightful comments and suggestions.  We have addressed these points as described below, and feel that the manuscript has been improved by their incorporation.

Response to reviewer 2 comments:

  1. The figure presented in the manuscript is incorrectly numbered. Section 4.2 describes kink-turn module, but the figure number shown at lines 280, 285 and 292 is Figure 3C which represents the complex with Fab antibody. These should be Figure 3B. Figure 3C seems to be for section 4.3. The crystal structure represented in Figure 3C seems to be cited in Reference 81 according to the PDB, but Reference 81 does not appear in Section 4.3.

We thank the reviewer for their careful reading of the manuscript and have changed the figure citations to match the figures (line 288-309) and included a citation for the structure in Figure 3C (line 329).

  1. References 53 through 115 seem to be off by one because the reference following “Cate et al.” in line 206 should be the reference 114. Please revise correctly.

We have fixed the references to represent the correct sources in the article (now reference 61).

  1. In line 401, it is better to insert “anomalous” before diffraction.

We have added anomalous here as suggested to avoid misunderstanding (line 431).